# The Therapeutic Strategies for Uremic Toxins Control in Chronic Kidney Disease

**DOI:** 10.3390/toxins13080573

**Published:** 2021-08-17

**Authors:** Ping-Hsun Lu, Min-Chien Yu, Meng-Jiun Wei, Ko-Lin Kuo

**Affiliations:** 1Department of Chinese Medicine, Taipei Tzu Chi Hospital, Buddhist Tzu Chi Medical Foundation, New Taipei 23142, Taiwan; 101318121@gms.tcu.edu.tw (P.-H.L.); yu7777c@gms.tcu.edu.tw (M.-C.Y.); u105030061@cmu.edu.tw (M.-J.W.); 2School of Post-Baccalaureate Chinese Medicine, Tzu Chi University, Hualien 97048, Taiwan; 3Division of Nephrology, Taipei Tzu Chi Hospital, Buddhist Tzu Chi Medical Foundation, New Taipei 23142, Taiwan; 4School of Medicine, Buddhist Tzu Chi University, Hualien 97048, Taiwan

**Keywords:** chronic kidney disease, diet control, dietary supplement, complementary and alternative medicine, uremic toxin, conventional medical therapy

## Abstract

Uremic toxins (UTs) are mainly produced by protein metabolized by the intestinal microbiota and converted in the liver or by mitochondria or other enzymes. The accumulation of UTs can damage the intestinal barrier integrity and cause vascular damage and progressive kidney damage. Together, these factors lead to metabolic imbalances, which in turn increase oxidative stress and inflammation and then produce uremia that affects many organs and causes diseases including renal fibrosis, vascular disease, and renal osteodystrophy. This article is based on the theory of the intestinal–renal axis, from bench to bedside, and it discusses nonextracorporeal therapies for UTs, which are classified into three categories: medication, diet and supplement therapy, and complementary and alternative medicine (CAM) and other therapies. The effects of medications such as AST-120 and meclofenamate are described. Diet and supplement therapies include plant-based diet, very low-protein diet, probiotics, prebiotics, synbiotics, and nutraceuticals. The research status of Chinese herbal medicine is discussed for CAM and other therapies. This review can provide some treatment recommendations for the reduction of UTs in patients with chronic kidney disease.

## 1. Introduction

Chronic kidney disease (CKD) is characterized by a gradual decrease in the glomerular filtration rate and proteinuria. CKD is a global health problem, and its incidence has been increasing. The estimated global prevalence of CKD is 8–14% [1,2]. When kidney function deteriorates gradually, many metabolites accumulate in the body. These accumulated substances, termed as uremic toxins (UTs), can result in adverse pathophysiological outcomes [3]. UTs can affect multiple organs and cause renal fibrosis, vascular calcification, anemia, peripheral arterial disease [4], adynamic bone disease [5], adipocyte dysfunction with insulin resistance [6], impaired immune system [7], uremic pruritus [8], and impaired valsartan-induced neovascularization [9].

In 1999, Vanholder and other researchers established the European Uremic Toxin (EUTox) Work Group and divided UTs into three categories based on their solubility, molecular weight, and ability to bind to serum proteins. These categories are small water-soluble compounds such as creatinine and uric acid (UA), middle molecules such as tumor necrosis factor (TNF)-α and interleukin (IL)-1β, and compounds that bind to proteins such as indoxyl sulfate (IS) [10]. The EUTox Work Group continues to identify new UTs and specify their standard and uremic concentrations [3]. Esmeralda et al. revised some UTs items based on several inflammatory markers that would increase the deterioration of CKD [11]. Currently, many hundreds of UTs including small water-soluble solutes, which can be removed through dialysis, and larger and protein-bound molecules, which are less likely to be removed during dialysis [12]. The pathophysiological mechanisms through which UTs cause multiple organ damage are complex and not completely understood. These mechanisms may include inflammation, reactive oxidative stress, cellular transdifferentiation, impaired mitochondria function, intestinal barrier destruction, and changes in intestinal microbiota [12,13,14]. Figure 1 shows the proposed mechanism of UTs generation. After food enters the intestine, it is not only digested by exocrine glands such as the liver, gallbladder, intestines, and pancreas but also decomposed by intestinal microbiota. Protein catabolism produces amino acids. If many ingredients that are favored by harmful intestinal microbiota are consumed, including tryptophan and tyrosine, then they will be converted into UTs through the proteolysis of intestinal microbiota [13,15]. Precursors such as indole and p-cresol are absorbed by human intestinal villi cells into the portal blood circulation of the liver. Subsequently, the liver metabolizes them into IS and p-tolyl sulfate [16]. CKD patients are more prone to constipation than ordinary people. The retention of feces will change the intestinal microbiota and promote the production and accumulation of more UTs. When the anaerobic bacteria that can produce short-chain fatty acids (SCFAs) are reduced, SCFAs such as butyrate and acetate, that promote intestinal contraction are also reduced, which in turn makes constipation worse [17,18]. The accumulation of UTs destroys the protective barrier of the intestinal epithelium, leading to the transfer of microbiota from the intestine to the body [13,19]. After IS produced in the liver enters the blood circulatory system, which is absorbed by proximal renal tubular cells through organic anion transporters (OAT) 1 and OAT3 at the basolateral membrane. The accumulated IS induces oxidative stress and then activates nuclear factor E2-related factor 2 (Nrf2) [20]. Some of these UTs can destroy mitochondrial function. Mitochondria may also be the source of UTs because mitochondria contain certain enzymes involved in the UTs synthesis pathway; thus, they may be the site of UTs synthesis. Moreover, because mitochondria are the key regulators of cellular redox homeostasis, they may directly affect the production of UTs. In addition, because many metabolites can be degraded in mitochondria, mitochondrial dysfunction might cause the accumulation of this toxin in organisms. Therefore, CKD can lead to the appearance of compounds that destroy mitochondria, and subsequent mitochondrial damage can cause further accumulation of UTs; the synthesis of UTs is related to mitochondria [14]. CKD reduces the ability to excrete UTs, leading to the accumulation of UTs in blood. At the same time, UTs can accelerate the deterioration of kidney function, leading to a vicious circle. All these factors together lead to the typical destruction of normal metabolic balance and uremic homeostasis that results in inflammation and uremia, causing multiple organ damage [12]. According to the review articles, due to the use of a highly permeable membrane with a greater pores radius and better preservation of the residual renal function, peritoneal dialysis (PD) could be anticipated that some uremic toxins are more efficiently cleared across the peritoneal membrane [21], and that the plasma levels of p-Cresol (protein-bound uremic toxin) are lower than in hemodialysis patients [22].

This review describes the potential nonextracorporeal methods from bench to bedside that can be used to reduce UTs levels and improve renal function based on the aforementioned mechanisms. The PubMed, Embase, Cochrane Library, Chinese National Knowledge Infrastructure, Airiti library, and Wanfang databases were searched by using the term “uremic toxin”. To broaden the search, we further reviewed the included articles and citations utilizing the “related articles” facility on PubMed. This paper is organized as follows. The first section focuses on medications including intestinal sorbents, UA modulators, and enzyme inhibitors. The second section focuses on diet control and supplement therapies such as probiotics, prebiotics, synbiotics, and nutraceuticals. The final section describes complementary and alternative medicine (CAM) and other therapies.

## 2. Conventional Medication Therapy

In conventional medical therapies, the serum concentration of UTs is mainly reduced using drugs that control underlying diseases to slow down the deterioration of the kidneys, like decrease hypertension, neutralize catecholamines, combat fluid overload, combat dyslipidemia and anemia [23], exert antioxidative effects [24], cause the metabolic degradation of UTs [25], change the bacterial amino acid metabolism [26], inhibit UA synthesis [27], inhibit sulfotransferase [28], reduce amino acid degradation [29], inhibit renal OAT [30], and combine or reduce the intake of toxins or their precursors to adjust the intestinal absorption capacity [31]. In addition, dialysis is performed for extracorporeal removal of UTs; however, the removal of middle and protein-bound UTs through conventional dialysis is inadequate [12]. The following is a brief discussion of treatment strategies that can be used to reduce UTs levels (Table 1).

### 2.1. Acarbose

Acarbose is a small intestinal alpha-glucosidase inhibitor that can increase the number of undigested carbohydrates reaching the colon, increase the production of short-chain fatty acid butyrate in the colon, and reduce intraluminal power of hydrogen (pH) value, thereby inhibiting bacterial deamination and increasing the use of ammonia. In a clinical trial, nine healthy people were treated with acarbose for 3 weeks, and their serum p-cresol declined significantly through changes in bacterial amino acid metabolism [26].

### 2.2. AST-120

AST-120 can absorb UTs and their precursors in the gastrointestinal tract and then excrete them in feces, reducing the absorption of UTs into the blood. For example, indole is produced through tryptophan metabolism by bacteria in the gastrointestinal tract. AST-120 absorbs indole and reduces its conversion into IS [31]. In an adenine-induced CKD mouse study, Sato et al. demonstrated that the administration of AST-120 led to a decline in accumulated IS and p-cresol sulfate (PCS) in multiple organs, such as the kidneys, heart, brain, and skeletal muscles [37]. The effect of AST-120 on renal disease progression had been reported in some large-scale clinical trials. The Carbonaceous Oral Adsorbent’s Effects on Progression of Chronic Kidney Disease (CAP-KD) trial demonstrated administration of AST-120 can significantly suppress the decrease in estimated glomerular filtration rate (eGFR) for a short observation period of one year [40]. However, it failed to suppress the composite endpoint consisting of doubling of serum creatinine (Scr), Scr ≥ 6.0 mg/dL, end-stage renal disease, and death. In Evaluating Prevention of Progression in CKD (EPPIC)-1 and EPPIC-2 trails, AST-120 was administered to 2035 adults with moderate to severe CKD. AST-120 administration was unable to suppress the primary endpoint which is the initiation of dialysis, transplantation, and doubling of Scr levels [32]. However, the secondary endpoint, the eGFR decline rate was significantly suppressed. The Kremezin study against renal disease progression in Korea (K-STAR) trial included measurement of plasma IS concentration. The primary endpoint was not suppressed in the AST-120 group [41]. Similarly, the eGFR decline tended to be suppressed. Finally, a meta-analysis including eight studies revealed that AST-120 can effectively lower IS levels but still controversial in terms of slowing disease progression and all-cause mortality [32]. Additional large-scale randomized controlled trials (RCTs) with longer follow-up durations and standardized outcomes are still necessary to clarify the clinical efficacy of AST-120.

### 2.3. L-Carnitine

Patients undergoing hemodialysis (HD) may develop L-carnitine deficiency due to intestinal malabsorption, reduced carnitine synthesis, and reduced carnitine clearance during dialysis. In a double-blind, placebo-controlled, crossover trial of patients undergoing HD, supplementation of L-carnitine, an antioxidant, for 8 weeks reduced the malondialdehyde (MDA) level, increased reduced/oxidized glutathione, and enhanced glutathione peroxidase activity and protein carbonyl concentration without adverse clinical effects [33]. The intravenous injection of carnitine supplementation has been shown to have a nephroprotective effect on contrast-medium nephropathy in an open-label, crossover study, it can reduce the neutrophil gelatinase associated lipocalin and SCr [42]. An animal study suggested that the nephroprotective effect of L-carnitine could reduce blood urea nitrogen (BUN), Scr, and MDA levels in CKD rats following right nephrectomy, possibly through antioxidative properties and free radical scavenging [38].

### 2.4. Cilastatin

Imipenem is a carbapenem antibiotic with nephrotoxicity. However, cilastatin, the renal dehydropeptidase-I (DHP-1) inhibitor, can prevent imipenem-induced nephrotoxicity by inhibiting the OAT-mediated transport of imipenem [43]. Cilastatin or probenecid ameliorated kidney injury and reduced BUN, Scr, and the renal secretion of imipenem in an imipenem-induced nephrotoxicity rabbit model through inhibition of renal OATs [30].

### 2.5. Cyclosporine A

Cyclosporine A (CsA), a calcineurin inhibitor, is used as an immunosuppressant. Although CsA is considered to be a nephrotoxic agent, its single injection can protect the kidneys during ischemia and reperfusion. In an ischemia-reperfusion (I/R) renal injury mice model, Lemoine et al. demonstrated that the administration of CsA improved mitochondrial respiration and reduced BUN and Scr probably through the inhibition of mitochondrial permeability transition [24].

### 2.6. Enalapril

Enalapril, an angiotensin-converting enzyme inhibitor, is used to treat cardiovascular diseases. Marek et al. found that in Wistar rats, enalapril treatment reduced the plasma trimethylamine-N-oxide (TMAO) level by increasing glomerular filtration and urine output; however, it did not reduce the plasma level of IS or change the bacterial composition in the gut [39].

### 2.7. Folate and Methylcobalamin

Folic acid (FA) is the main cofactor of total homocysteine (Hcy) metabolism, and methylcobalamin (Me-Cbl) is the methylated form of cobalamin. Both play key roles in the conversion of Hcy to methionine. The combined administration of oral FA and intravenous Me-Cbl can reduce plasma Hcy in HD patients [44], which may prevent cardiovascular disease and stroke [45]. In a prospective RCT of patients undergoing HD, both oral FA supplementation and intravenous (i.v.) injection of Me-Cbl plus oral FA normalized plasma total Hcy; however, i.v. injection of Me-Cbl showed no change [34]. Another RCT reported that patients undergoing HD who received i.v. injection of Me-Cbl plus oral FA showed a reduction in Hcy and asymmetric dimethylarginine (ADMA) levels; this reduction possibly resulted from an increase in the metabolic degradation of UTs [25].

### 2.8. Ketoacids

Ketoanalogues have been used in chronic kidney disease for many years. It can combine the excess amino acids to synthesize essential amino acids (EAAs) through transamination, which can reduce the formation of nitrogenous waste and urinary toxins, and reduce the degradation of protein, which in turn, can improve the patient’s renal function and prognosis. The transamination of ketoanalogues is bidirectional and is related to the concentration of amino acids. When the concentration of amino acids in the body was high, the body will tend to decompose ketonanalogues instead of synthesizing them into new EAAs [46]. In a pilot study published in 2013, researchers analyzed the physiological data of 32 patients with stage 3 CKD. These patients were randomly divided into two groups: very low protein diet (VLPD) with ketoacid (KA) analog supplementation for 1 week, followed by low protein diet (LPD) administration in the second week, and LPD administration for 1 week, followed by VLPD with KA analog supplementation in the second week. The results indicated that restricted protein consumption with KA analog supplementation reduced the production and intestinal absorption of IS [35]. In another clinical trial, 17 patients were included and randomly divided into two groups: LPD and VLPD with EAAs and KA supplementation. After the 24-week trial, forearm protein metabolism was affected in both groups. Overall, skeletal muscle may adapt to restricted protein intake by combining the responses of decreased protein degradation and increased amino acid recycling [29]. 

### 2.9. Meclofenamate

Meclofenamate is a type of nonsteroidal anti-inflammatory drug (NSAID) that inhibits hepatic IS production. Saigo et al. used an I/R rat kidney model to demonstrate that meclofenamate considerably improved renal function and reduced BUN, Scr, and IS. In addition, they indicated that meclofenamate could inhibit sulfotransferase (SULT) and reverse the downregulation of rat OAT1 and rat OAT3 expression by preventing prostaglandin E2 generation [28].

### 2.10. Reduced Glutathione

Oxidative stress arises when there is an imbalance between free radical production and antioxidant defense. Uremic toxins can be a source of oxidative stress in CKD patients. Retention of these toxins promotes systemic inflammation via priming polymorphonuclear leukocytes and stimulating CD-8+ cells [47]. Additional associative factors of oxidative stress in CKD include low serum selenium concentration, low platelet glutathione peroxidase activity [48], and lower serum levels of glutathione [48]. Although glutathione is critical to fight against oxidative stress, some evidences disclosed that glutathione is poorly absorbed by oral route mainly due to the action of an intestinal enzyme, the γ-glutamyl transpeptidase which degrades glutathione [49]. Wang et al. ever found that the oral supplementation of reduced glutathione in patients undergoing HD significantly reduced their IL-6 and TNF-α levels but not their Scr or BUN levels in their RCT trial [36]. Although the very low oral bioavailability limits the interest of oral glutathione supplementation, Schmitt et al. demonstrated the superiority of a new sublingual form of glutathione over the oral form in terms of glutathione supplementation [50]. A new sublingual form may be a promising antioxidant therapy in CKD patients in the future.

## 3. Diet Control and Diet Supplements

UTs are usually produced through the metabolism of amino acids by intestinal microbiota microbiota, and accumulated UTs can change the composition of intestinal microbiota. Increasing evidence shows that intestinal microbiota plays a crucial role in the development of CKD [51]. Therefore, the current methods of reducing UTs through diet control and supplement therapy include reducing protein intake through a low-protein or plant-based diet [52] and supplementing probiotics, prebiotics, and synbiotics to change the composition of intestinal microbiota; some supplements could exert anti-inflammatory and antioxidative effects [53]. These treatment strategies are briefly discussed below (Table 2).

### 3.1. Diet^®^ k/d^®^

Prescription Diet^®^ k/d^®^, an antioxidant-enriched food, includes the supplementation of alpha-linolenic acid (ALA) (18:3 [n-3]) to manage CKD in aging dogs. Dogs with International Renal Interest Society (IRIS) Stage 1 CKD fed with Prescription Diet k/d for 12 months exhibited decreased BUN, Scr, and symmetric dimethylarginine (SDMA) [60].

### 3.2. Lingonberry

Lingonberry (*Vaccinium vitis-idaea* L.) is an evergreen dwarf shrub rich in anthocyanins, which has anti-inflammatory, anti-diabetic, and renal protection properties [64,65]. In another study, mice were fed with a high-fat diet (HFD) to induce chronic kidney injury for investigating the effects of lingonberry on CKD. They found that the serum levels of inflammatory cytokines such as TNF-α and IL-6 decreased and renal injury improved in the HFD group that received lingonberry supplementation. In addition, lingonberry supplementation attenuated palmitic acid-induced nuclear factor-kappa B (NF-κB) activation and inflammatory cytokine expression in proximal tubule epithelial cells. Lingonberry supplementation may protect the kidney from metabolic abnormality-induced injury by preventing an inflammatory response [61].

### 3.3. Mitoquinone

Mitoquinone (MitoQ), a synthesized mitochondrially targeted antioxidant that contains a lipophilic triphenylphosphonium cation and coenzyme Q10, can block reactive oxygen species (ROS) and prevent mitochondrial oxidative damage which was applied to treat various diseases including liver fibrosis and neurodegenerative disease [66,67]. MitoQ reduced Scr, IL-1β, IL-6, and TNF-α levels in an I/R CKD mouse model. Moreover, MitoQ recovered intestinal barrier dysfunction by ameliorating mitochondrial deoxyribonucleic acid (DNA) damage [62].

### 3.4. Prebiotic Oligofructose-Enriched Inulin

Oligofructose, a type of oligosaccharide fructans, was used as an alternative sweetener. Probiotic bacteria such as *Bifidobacteria* and *Lactobacilli* are saccharolytic in nature. They cause oligofructose to produce SCFAs that exert a positive immunomodulatory effect, maintaining the symbiosis of intestinal microbiota [68]. In a nonrandomized, open-label trial of patients undergoing HD, administration of prebiotics composed of prebiotic oligofructose-enriched inulin (OF-IN) for 4 weeks reduced the generation rate and serum concentration of PCS but not that of IS. Another study proposed that prebiotics possessed the ability to modify intestinal uremic metabolites [54].

### 3.5. Probiotics

Probiotics were defined as “live microorganisms” by the World Health Organization (WHO) and Food and Agriculture Organization in 2002. Emerging evidence suggests that an appropriate supplementation of probiotics can improve gastrointestinal health, strengthen the immune system, lower the incidence of allergies, alleviate the symptoms of lactose intolerance, and reduce the risks of cancer [69,70]. The mechanisms by which probiotics provide their effects may involve modifying the pH of the intestine, defeating pathogens by producing antibacterial compounds, competitively excluding pathogens at the binding site, and combining of harmful mutagens and carcinogens [71]. Many studies have suggested that oral supplementation of a probiotic formula of selected microorganisms may exert a nephroprotective effect by extracting UTs in patients with CKD. In an RCT, the administration of Kibow Biotics probiotic formulation (*L. acidophilus* KB27, *B. longum* KB31, and *S. thermophilus* KB19) in patients with CKD reduced BUN, Scr, and UA levels; enhanced patients’ quality of life; and resulted in no adverse effects [55]. Another study demonstrated that the administration of a probiotic containing Bifidobacterium longum in gastric-resistant seamless capsules could effectively reduce serum Hcy and IS levels in patients with HD [56].

### 3.6. Short-Chain Fatty Acids

In CKD and end-stage renal disease (ESRD), gut-derived UTs can cause systemic inflammation, oxidative stress, and cause dysbiosis of the gut. When the anaerobic bacteria that can produce SCFA declines, resulting in insufficient production of SCFA [72]. SCFAs was mainly composed of acetate, propionate, and butyrate and exerted a positive immunomodulatory, anti-inflammatory, anti-oxidative, antibacterial, and anti-diabetic effects effect [73,74,75]. In a nonrandomized pilot study, the food additive sodium propionate (SP), an SCFA, was administered in the form of capsules, and this effectively reduced IS, MDA, and PCS in patients undergoing HD. Moreover, SP reduced pro-inflammatory parameters and oxidative stress and improved insulin resistance and iron metabolism [57].

### 3.7. Soluble Fiber and Omega-3 Fatty Acids

Omega-3 fatty acids (O3FAs) are a group of polyunsaturated fatty acids, among which eicosapentaenoic acid (EPA), ALA, and docosahexaenoic acid (DHA) are closely related to human physiology. O3FAs can be gained primarily from vegetable seed oils or aquatic organisms [76]. O3FA plays a crucial role in the eicosanoid biosynthesis. EPA competes with arachidonic acid to switch the synthesis pathway to promote the production of anti-inflammatory eicosanoids [77]. In addition, eicosanoids derived from O3FA have shown clinical benefits for cardiovascular disease, diabetes, and nephropathies due to their anti-inflammatory effects [77,78].

Omega-3 supplementation could improve oxidative stress in patients with CKD [79]. Eating fiber-rich foods can reduce the risk of kidney disease, inflammation, and death [80]. Soluble fiber and omega-3 fatty acids may reduce IS, PCS, and SDMA by modulating the gut microbiota [63].

### 3.8. Synbiotic

Synbiotics, a combination of probiotics and prebiotics, has immune system regulation and anti-inflammatory effects. It can improve the intestinal environment and reduce concentration of nitrogen-containing metabolites [81,82], was applied to treat intestinal chronic diseases [82], allergic rhinitis and asthma [83]. An uncontrolled trial showed that a synbiotic consisting of *Lactobacillus casei*, *Bifidobacterium breve*, and galactooligosaccharides reduced the PCS level but did not change IS or phenol levels in in patients undergoing HD [58].

### 3.9. Vegetarian Diet

Plants are the dietary source of fibers, which shifts the gut microbiota to increased generation of anti-inflammatory compounds and reduced production of uremic toxins [84]. Vegetarian diets have a low endogenous acid load, which could reduce metabolic acidosis in CKD patients and potentially slow CKD progression [84,85]. Moreover, in the primary prevention of CKD, our previous study showed that vegetarian diets a strong negative association between vegetarian diets and the prevalence of CKD [86]. 

Another review supports this observation and revealed that plant-based foods and Mediterranean (MD) diet or Dietary Approaches to Stop Hypertension (DASH) diets could reduce many types of UTs; however, the potential risk of hyperkaliemia should be examined [87]. A cohort study revealed that vegetarian diet patients in hemodiafiltration have not only lower plasma levels of IS and PCS, but also lower serum level of urea [59].

### 3.10. Vitamin D

CKD patients are prone to vitamin D deficiency, which can cause a variety of diseases. Higher 25-hydroxyvitamin D (25[OH]D) levels can significantly improve the survival rate of CKD patients [88].

Case–control studies have shown that vitamin D supplementation could improve endothelial dysfunction in patients with CKD [89]. An RCT reported that vitamin D supplementation for 8 weeks could reduce the IL-6 and UA levels in patients with nondiabetic CKD stage 3–4 and vitamin D deficiency. In addition, vitamin D supplementation could improve vascular function, as determined by significant positive changes in pulse wave velocity [53].

## 4. Complementary and Alternative Medicine Therapy

CAM therapy is widely used in CKD treatment. A meta-analysis including 20 RCTs showed that in patients with diabetic kidney disease (DKD), traditional Chinese medicine (TCM) as an adjunctive treatment significantly reduced BUN and Scr levels, and improved uremic protein excretion and quality of life compared to the placebo group with less adverse events [90]. The clinical data of high-quality CAM therapy is indeed scarce. Prescriptions and acupuncture have been studied in RCT grade [91,92], and the evidence of moxibustion belongs to meta-analysis grade [93]. However, most of the CAM therapy studies are still animal experiments. Fortunately, it is indeed observed that CAM therapy has the effect of reducing urinary toxins and improving renal function in animal experiments [94]. Increasing evidence has revealed that CAM therapy can slow the progression of renal damage through mechanisms such as improving renal microcirculation, exerting anti-inflammatory and antioxidative effects, and inhibiting programmed cell death [94]. Therefore, this section briefly describes the nephroprotective effects of CAM therapies and their ability to control UTs. CAM therapeutic methods applied for CKD include TCM preparations, herbal active principle, acupuncture, and moxibustion (Table 3).

### 4.1. Curcuma Longa and Boswellia Serrata

Dietary turmeric, rich in curcumin, has anti-inflammatory, anti-oxidant, and chemotherapeutic effects [117]. Curcumin has been shown to reduce inflammation in patients with type 2 diabetes [118]. CKD rat models have confirmed the anti-inflammatory effects of curcumin [119], and *Boswellia serrata* could improve the inflammatory symptoms of colitis and knee osteoarthritis [120,121]; however, their efficacy in patients with CKD has not been examined. An RCT including patients with CKD revealed a significant decrease in plasma IL-6 following treatment with oral *C. longa* and *B. serrata* for 8 weeks compared with a placebo group; however, other variables such as TNF-α and CRP exhibited no significant differences. Moreillon et al. indicated that *C. longa* and *B. serrata* reduce IL-6 possibly through inhibition of the pathways of NF-κB and mitogen-activated protein kinase [91].

### 4.2. Dahuang Fuzi Decoction

Dahuang Fuzi Decoction (DFD), a well-known traditional Chinese prescription, consists Dahuang (*Radix et Rhizoma Rhei*), Paofuzi (*Radix Aconiti Lateralis Preparata*), and Xixin (*Radix et Rhizoma Asari*), originated from the Synopsis of Golden Chamber [96]. DFD is often used to treat gynecological diseases and CKD [122], it could also reduce BUN, Scr, and UA levels in an adenine-induced renal injury rat model. Moreover, DFD could block the activation of transforming growth factor beta 1 c-Jun N-terminal kinase (TGF-b1-JNK) pathways to mitigate renal damage and tubular epithelial apoptosis [96].

### 4.3. Danhong Injection and Salvianolic Acids

Danhong injection (DHI) contains the extraction of two Chinese medicines, the radix and rhizome of *Salvia miltiorrhiza* Bunge (Labiatae) and the flower of *Carthamus tinctorius* L. (Asteraceae) [123]. Salvianolic acid B, rosmarinic acid, and lithospermic acid are powerful protein-binding ligands, and DHI and salvianolic acids are rich in these substances [123]. In a nephrectomized CKD rat model, injection of DHI and its salvianolic acids through the caudal vein increased the dialysis removal of IS and PCS. DHI and its salvianolic acids showed favorable ability as protein-bound competitors [97].

### 4.4. Uremic Clearance Granule

Uremic clearance granule (UCG), also called as NiaoDuQing (NDQ), created based on the TCM theory consisting of 16 herbs, such as *Rheum officinale*, *Glycyrrhiza uralensis*, *Astragalus membranaceus*, *Poria cocos*, *Sophora flavescens*, *Chrysanthemum morifolium*, and so forth, is the first Chinese medicine approved by the China Food and Drug Administration for the treatment of CKD. UCG has been clinically demonstrated to slow CKD progression [124]. Its effects include improving systemic micro-inflammation [125], reducing Scr [95], and calibrating calcium and phosphorus metabolic disorders [126]. However, its mechanism has not been thoroughly studied [95]. A multicenter double-blind, placebo-controlled, and randomized clinical trial recruited 292 patients with stage 3b-4 CKD proved that UCG could delay CKD progression safely and effectively [95]. In an adenine and unilateral ureteral obstruction (UUO)-induced CKD rat model, Huang et al. reported that the administration of UCG reduced BUN, Scr, and UA levels. In addition, UCG exhibited antifibrosis ability through regulation of extracellular matrix (ECM) degradation and associated signaling pathway activity [98].

### 4.5. Zhibai Dihuang Wan

Zhibai Dihuang Wan (ZDW) is a TCM preparation consisting of *Cornus officinalis* Siebold and Zucc, *Rehmannia glutinosa* (Gaertn.) DC., root, baked, *Dioscorea oppositifolia* L., *Phellodendron amurense* Rupr., bark, *Anemarrhena asphodeloides* Bunge, rhizome, *Paeonia suffruticosa* Andrews, root bark, *Alisma plantago-aquatica* L., rhizome and *Poria cocos* (Schw.)Wolf, that has been used to treat CKD and diabetes for thousands of years [127]. ZDW reduced BUN and Scr levels in rats with gentamicin-induced renal injury and exerted a protective effect on renal tubular cells through antiapoptotic effects by limiting caspase-3 activation [99]. Moreover, a study of aristolochic acid (AA)-intoxicated zebrafish demonstrated that ZDW treatment could attenuate AA-induced kidney malformations and suppress the expression levels of TNF-α [100].

### 4.6. Catechin Combined with Vitamin C and Vitamin E

Catechin, a type of flavonoid, is found in green tea and other plant foods, which has the benefit of protecting against congestive heart failure and reducing the incidence of myocardial ischemia [128,129]. Catechin, vitamin C, and vitamin E all have the favorable antioxidant capacity [130,131,132]. A study performed 5/6 nephrectomy in aged Wistar rats to explore the mechanism through which catechin protects against renal dysfunction. Catechin with vitamin C and E supplementation could reduce oxidative stress resulting from aging and renal failure and prevent ADMA accumulation [101].

### 4.7. Cyanidin-3-O-Glucoside

Anthocyanins are polyphenol compounds that are abundant in blueberries and purple corn [133], which can reduce the risk of cardiovascular disease-related mortality [134]. Through in vitro and in vivo investigations, Qin et al. found that cyanidin-3-O-glucoside (C3G), the most widespread anthocyanin, acts as an antioxidative agent and regulates glutathione metabolism, resulting in the reduction of BUN, Scr, and urine albumin–creatinine ratio (UACR) and the amelioration of pathological changes in kidney biopsy in db/db mice [102].

### 4.8. Epigallocatechin-3-Gallate

Green tea is rich in a variety of catechin polyphenols, which has anti-inflammatory, anti-oxidant, anti-viral, and anti-cancer effects [135,136,137]. Epigallocatechin-3-gallate (EGCG), which is extracted from green tea, is the richest and the most active catechin polyphenol. A UUO-induced CKD mouse study suggested that EGCG exerts anti-inflammatory and antioxidative effects by inhibiting the NF-κB signaling pathway and activating the nuclear factor E2-related factor 2-Kelch-like ECH-associated protein 1 (Nrf2-Keap1) pathway [103].

### 4.9. Gypenoside

*Gynostemma pentaphyllum* is a widely used and safe TCM. Recent studies have shown that it has anti-cancer, cardioprotective, anti-diabetic, and anti-inflammatory activities [138]. Gypenoside (GP), a major component of *Gynostemma pentaphyllum*, exerts anti-inflammatory and antioxidative effects [139]. However, whether GP can attenuate CKD remains unclear. In mice with renal I/R injury, intravenous GP injection reduced BUN, Scr, TNF-α IL-1β, IL-6, and MDA by inhibiting extracellular signal-regulated kinase (ERK) signaling [104].

### 4.10. Huangkui Capsule

Huangkui capsule, which is prepared using the extract of the flower *Abelmoschus manihot* (Linn.) Medicus (*A. manihot*), was approved by China’s State Food and Drug Administration in 1999 for treating chronic nephritis. Studies have confirmed the renal protective effect of the Huangkui capsule in vivo, in vitro, and clinically [105,140]. Flavonoids, the main chemical component of the Huangkui capsule, can be converted into glucuronic acid-sulfate conjugates in vivo, which may contribute to the nephroprotective effects [141]. A multicenter RCT showed that Huangkui capsule has antiproteinuric effect for patients with early stage primary glomerulonephritis, and the other multicenter, double-blind, double-dummy RCT also showed that it can effectively reduce proteinuria in patients with IgA nephropathy. Both RCTs showed that Huangkui capsule treatment has no obvious side effects [142,143]. Cai et al. demonstrated that the Huangkui capsule reduced BUN and Scr in adenine-induced chronic renal failure (CRF) rats. It exerted a preventive effect on tubulointerstitial fibrosis, which participates in the mechanism that inhibits the nicotinamide adenine dinucleotide phosphate oxidase (NADPH)/ROS/ERK pathway [105]. Another 5/6 nephrectomy rat model study revealed that the Huangkui capsule inhibited the synthesis of indole by gut bacteria by interfering with the transport of tryptophan, thereby effectively inhibiting the accumulation of IS in CKD rat [106].

### 4.11. Leonurine

Leonurus cardiac has antibacterial, antioxidant, and anti-inflammatory activities. For thousands of years, it has been traditionally used in China for gynecological diseases such as uterotonic action and postpartum blood stasis. Leonurus cardiac has been used in modern times to treat nervous system dysfunction and cardiovascular disease [144]. Leonurine (LEO), an alkaloid isolated from *Leonurus cardiac*, exerts antioxidative effects [145]. LEO reduced BUN and Scr levels and downregulated TNF-α, IL-1, IL-6, IL-8, and kidney injury molecule-1 (KIM-1) expression by inhibiting the ROS-mediated NF-kB signaling pathway in a mouse model of lipopolysaccharide (LPS)-induced renal injury [107].

### 4.12. Ligustrazine

Ligustrazine (LIG), an alkaloid extracted from *Ligusticum wallichii*, could reduce I/R-induced hepatic and endothelial cell damage by scavenging cytotoxic oxygen free radicals [146,147]. LIG reduced MDA and TNF-α levels in I/R-induced renal injury mice by downregulating oxidative stress and apoptosis and reducing neutrophil infiltration [108].

### 4.13. Notoginsenoside R1

*Panax notoginseng* is a TCM that promotes blood circulation and removes blood stasis. It is widely used in China to treat cardiovascular diseases. A study reported that notoginsenoside R1 (NR1) is the main component of *Panax notoginseng* that possesses anti-inflammatory and anticoagulant properties [148]. NR1 mitigated I/R-induced kidney dysfunction as determined by the reduced levels of serum Cr and TNF-α, possibly through suppressing the p38/NF-kB pathway in a rat model of I/R-induced renal injury [109].

### 4.14. Osthole

Osthole isolated from *Cnidium moonnieri* (L.) Cussion, which is a coumarin derivative, protects cerebral artery occlusion injury by exerting anti-inflammatory effects [149]. In an I/R-induced renal injury model, Luo et al. demonstrated that the administration of osthole reduced TNF-α, IL-8, and IL-6 levels. Osthole prevents renal injury by suppressing the Janus kinase 2/signal transducer and activator of transcription 3 (JAK2/STAT3) signaling pathway and activating PI3K/Akt signaling pathway [110].

### 4.15. Paeoniflorin

Paeoniflorin (PF) is a bioactive monoterpene glycoside extracted from *Radix Paeoniae Rubra*, which could inhibit colitis by inhibiting the expression of Toll-like receptor 4 (TLR4) [150] and liver fibrosis by inhibiting macrophage activation [151]. PF reduced BUN, Scr, and IL-1β levels and attenuated inflammatory responses by inhibiting CXCR3/CXCL activation in mice with concanavalin A (ConA)-induced renal injury [111].

### 4.16. Resveratrol

Resveratrol (RSV), a natural polyphenol that originates in grapes, berries, and other dietary constituents, exerts antioxidant activity and anti-atherosclerotic effects by modulating the growth of certain gut microbiota, such as *Lactobacillus* and *Bifidobacterium* [152,153]. RSV can promote the release of nitric oxide and prostacyclin to maintain endothelial function and adjust vascular tone [154]. RCT confirmed that RSV has cardioprotective effects in patients with stable coronary artery disease [155]. Chen et al. reported that RSV attenuated TMAO in TMAO-induced atherosclerosis mice through the remodeling of gut microbiota [112].

### 4.17. Rhubarb

Rhubarb is a common herbal medicine composed of the dried rhizomes and roots of *Rheum palmatum* L, Rheum tanguticum Maxim that is broadly used in sepsis and pancreatitis to protect the intestinal mucosal barrier [156,157]. Rhubarb enema could alleviate IS overload to ameliorate renal tubulointerstitial fibrosis in the 5/6 nephrectomy renal injury rats model by reducing kidney oxidative stress and inflammatory damage [114]. Furthermore, Ji et al. reported that rhubarb enema could reduce IL-1β and IL-6 levels in 5/6 nephrectomy renal injury rats by modifying intestinal microbiota [115].

### 4.18. 10-(6′-Plastoquinonyl) Decylrhodamine 19 

Mitochondria play an important role in the pathogenesis of uremia, and their damage will lead to the accumulation of UTs [14]. The compartment of the mitochondria is negatively charged in the cell, the design of active molecules combined with cationic carriers can accumulate in mitochondria due to their positive charges [158]. 10-(6′-plastoquinonyl) decylrhodamine 19 (SkQR1), a mitochondrial-targeted antioxidant conjugated with plastoquinone and cationic decylrhodamine 19, possesses high antioxidant capacity in animal models and in vitro [159]. SkQR1 exerted renal protection by increasing erythropoietin levels in gentamicin-induced renal injury rats. In addition, SkQR1 could reduce BUN and MDA levels [116].

### 4.19. Tanshinone I

Danshen, also named *Salvia miltiorrhiza*, is a traditional Chinese medicine often used to treat cardiovascular diseases. Tanshinone I, an active component of *Salvia miltiorrhiza* Bunge, has the effects of anti-inflammation, cardiovascular protection, anti-tumor, and anti-hepatic fibrosis [160,161]. Tanshinone I exerted a renal protective effect, as indicated by reduced BUN and Scr levels, through the enhancement of AA metabolism by inducing CYP1A in a mouse model of AA-induced renal injury [113].

### 4.20. Acupuncture

Acupuncture is a popular CAM therapeutic method that can effectively treat certain diseases such as pain and insomnia; this method is recognized by WHO [162]. Acupuncture can stimulate the production of endomorphin-1, encephalin, β-endorphin, and serotonin in plasma and brain tissue, thereby achieving the effects of analgesia, sedation, and immune regulation [163]. In an RCT including 53 patients with CKD, once-weekly electroacupuncture at bilateral Hegu (LI4), Zusanli (ST36), and Taixi (KI3) for 12 weeks improved renal function; the reduction in Scr levels induced by reduced eGFR was greater in the acupuncture group than in the sham acupuncture group [92]. Acupuncture might improve renal function by regulating sympathetic nerves and activating biologically active chemicals [164].

### 4.21. Moxibustion

Moxibustion, a CAM therapy that consists of burning dried moxa on particular acupoints on the body, is used to treat a variety of diseases, such as malposition, diarrhea, soft tissue damage, and dysmenorrhea [165]. Moxibustion methods can be divided into direct moxibustion and indirect moxibustion. Direct moxibustion is to place moxa sticks on acupuncture points, while indirect moxibustion is defined as placing medicinal materials (e.g., mugwort, ginger, etc.) between moxa sticks and acupuncture points [166]. Moreover, the moxa sticks are placed on acupuncture needles inserted into the skin to improve the curative effect [167]. The pharmacological effects of moxibustion are the heat and light radiation of moxibustion burning and aromatherapy [168,169]. Moxibustion may improve CKD by exerting anti-inflammatory effects [170]. A meta-analysis including 23 RCTs revealed that in patients with CKD, moxibustion therapy significantly reduced BUN and Scr levels and improved uremic protein excretion and quality of life. In patients with CKD, moxibustion causes expansion of local renal capillaries and reduces renal podocyte damage [93].

## 5. Conclusions

Different treatments, such as medicine, diet control, diet supplement, and CAM, employ different mechanisms of action to improve kidney function or reduce different UTs, including the inhibition of inflammation, cell apoptosis, removal of toxins, regulation of intestinal bacteria, and inhibition of oxidative stress. This evidence supports the potential applications of these therapies. Growing evidence has revealed that the intestine is the main source of UTs. Regulating diet and modulating intestinal microbiota can reduce the accumulation of UTs. Recent studies have indicated that reducing the production of UTs in mitochondria can protect mitochondria, thus reducing the pathological consequences of CKD. Some Chinese medicines have shown the ability to protect the kidneys and reduce UTs; however, certain Chinese medicines are harmful to the kidneys and increased attention should be focused on the use of Chinese medicines. The scale of clinical trials on Chinese medicines is small, and many treatments are still in the animal experiment stage; however, additional large-scale trials are needed to examine their effects on reducing UTs or improving residual renal functions.

## Figures and Tables

**Figure 1 toxins-13-00573-f001:**
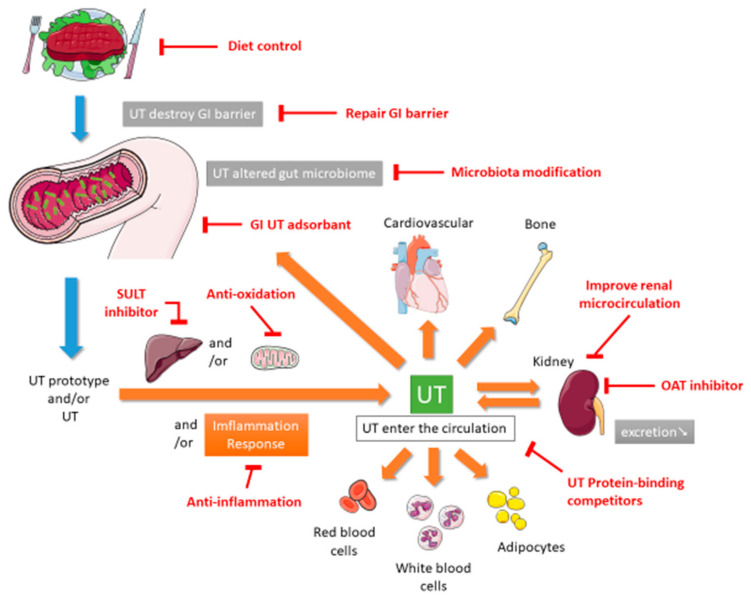
Proposed mechanism of UT generation and therapeutic methods. GI, gastrointestinal; OAT, organic anion transporter; SULT, sulfotransferase; UT, uremic toxin.

**Table 1 toxins-13-00573-t001:** Medications for the control of UTs.

Intervention	Route, Dosage and Frequency	Author/Year	Mechanism/Usage	Study Design	Subjects	Subject Number	Result
Clinical Studies							
Acarbose	Oral, 100 mg, TID	Evenepoel et al., 2006 [26]	Changes in bacterial amino acid metabolism	Clinical trial	Healthy people	9	PCS ↘
AST-120	Oral, 2.7 to 9 g/day	Chen et al., 2019 [32]	UT adsorbent	Meta-analysis	Patients with CKD	3349	IS ↘
L-carnitine	i.v., 20 mg/kg, 3 times/week	Fatouros et al., 2010 [33]	Antioxidation	Clinical trial	Patients undergoing HD	12	MDA ↘
Folate	Oral, 10 mg, QD	Trimarchi et al., 2002 [34]	Metabolic degradation of UT	RCT	Patients undergoing HD	62	Hcy ↘
Folate and Methylcobami	i.v. methylcobalami 500 µg, 3 times/week and oral folate 15 mg, QD	Koyama et al., 2010 [25]	Metabolic degradation of UT	RCT	Patients undergoing HDs	40	ADMA ↘, Hcy ↘
Ketoacid and LPD	Oral, 1 pill/5 kg, QD	Marzocco et al., 2013 [35]	Decreased amino acid degradation/protein carbamylation	RCT	CKD stage 3 adults	32	IS ↘
Ketoacid and LPD	Oral, 0.1 g/kg, TID	Garibotto et al., 2018 [29]	Decreased amino acid degradation/protein carbamylation	RCT	Patients with CKD	17	Urea ↘
Reduced glutathione	Oral, 400 mg, TID	Wang et al., 2016 [36]	Antioxidation	RCT	Patients undergoing HD	150	IL-6 ↘, TNF-α ↘
Animal Studies							
AST-120	Oral, 8% *w*/*w*, QD	Sato et al., 2017 [37]	UT adsorbent	Animal	Adenine-induced CKD mice	24	IS ↘, PCS ↘
L-carnitine	i.p., 500 mg/kg, QD	Sener et al., 2004 [38]	Antioxidation	Animal	Right nephrectomy rats	16	BUN ↘, Cr ↘, MDA ↘
Cilastatin	i.v., 200 mg/kg, once	Huo et al., 2019 [30]	OAT inhibitor	Animal	Imipenem-induced nephrotoxicity rabbits	4	BUN ↘, Cr ↘
cyclosporine	i.v., 3 mg/kg, once	Lemoine et al., 2015 [24]	Antioxidation	Animal	I/R mice	22	BUN ↘, Cr ↘
Enalapril	Oral, 12.6 mg/kg, QD	Marek et al., 2018 [39]	ACEI, increased glomerular filtration, and urine output	Animal	Wistar rats	27	TMAO ↘
meclofenamate	i.v., 10 mg/kg, TID	Saigo et al., 2014 [28]	SULT inhibitors	Animal	Renal I/R rats	9	BUN ↘, Cr ↘, IS ↘
Probenecid	i.v., 50 mg/kg, once	Huo et al., 2019, [30]	OAT inhibitor	Animal	Imipenem-induced nephrotoxicity rabbits	12	BUN ↘, Cr ↘

↘, decrease; ACEI, angiotensin converting enzyme inhibitor; ADMA, asymmetric dimethylarginine; BUN, blood urea nitrogen; CKD, chronic kidney disease; Cr, creatinine; Hcy, homocysteine; HD, hemodialysis; i.p., intraperitoneal; i.v., intravenous; I/R, ischemia/reperfusion; IL-6, interleukin-6; IS, indoxyl sulfate; LPD, low protein diet; MDA, malondialdehyde; OAT, organic anion transporter; PCS, p-cresyl sulfate; QD, quaque die; RCT, randomized controlled trial; SULT, sulfotransferase; TID, ter in die; TMAO, trimethylamine N-oxide; TNF-α, tumor necrosis factor alpha; UA, uric acid.

**Table 2 toxins-13-00573-t002:** Diet treatments for the control of UTs.

Intervention	Route, Dosage and Frequency	Author/Year	Mechanism/Usage	Study Design	Subjects	Subject Number	Result
Clinical Studies							
Prebiotics—OF-IN	Oral, 10 g, BID	Meijers et al., 2010 [54]	Modulating intestinal microbiota	Open-label phase I/II study	Patients undergoing HD	22	PCS ↘
Probiotics: *L. acidophilus* KB27, *B. longum* KB31, and *S. thermophilus* KB19	Oral, 2 capsules, TID	Ranganathan et al., 2010 [55]	Modulating intestinal microbiota	RCT	Patients with CKD stages 3 and 4	46	BUN ↘, Cr ↘, UA ↘
Probiotics: *B. longum*	Oral, 3–12 × 10^9^ CFU/day	Taki et al., 2005 [56]	Modulating intestinal microbiota	Case series	Patients undergoing HD	27	Hcy ↘, IS ↘
SCFA: sodium propionate	Oral, 1 g, QD	Marzocco et al., 2018 [57]	Anti-inflammation and antioxidation	Clinical trial	Patients undergoing HD	20	IS ↘, MDA ↘, PCS ↘
Synbiotic: *L. casei*, *B. breve*, and galactooligosaccharides	Oral, 1 pack, TID	Nakabayashi el et al., 2011 [58]	Modulating intestinal microbiota	Clinical trial or case series	Patients undergoing HD	9	p-Cresol ↘
Vegetarian	Oral	Kandouz et al., 2016 [59]	Improvement of metabolic acidosis, modification of intestinal microbiota	Cohort	Patients in hemodiafiltration	138	IS ↘, PCS ↘, Urea ↘
Vitamin D	Oral, 300,000 IU, QD	Kumar et al., 2017 [53]	Anti-inflammation	RCT	Patients with nondiabetic CKD and vitamin D deficiency	120	IL-6 ↘, UA ↘
Animal Studies							
Diet^®^ k/d^®^	Oral, 1.6 RER, QD	Hall et al., 2018 [60]	Anti-inflammation	Animal	CKD dogs	36	BUN ↘, Cr ↘, SDMA ↘
Lingonberry	Oral, 5% *w*/*w*, QD	Madduma Hewageet al., 2020 [61]	Anti-inflammation	Animal	HFD-induced kidney injury mice	30	BUN ↘, Cr ↘, IL-6 ↘, TNF-α ↘
MitoQ	i.v., 4 mg/kg, once	Hu et al., 2018 [62]	Antioxidation through reducing mitochondrial ROS	Animal	I/R mice	24	Cr ↘, IL-1β ↘, IL-6 ↘, TNF-α ↘
Soluble Fiber and Omega-3	Oral, 3666 kcal/kg, QD	Ephraim et al., 2020 [63]	Modulating intestinal microbiota	Animal	Dogs aged older than 7 years	36	phenolic UTs ↘, SDMA ↘

↘, decrease; BUN, blood urea nitrogen; BID, bis in die; CFU, colony-forming units; CKD, chronic kidney disease; Cr, creatinine; Hcy, homocysteine; HD, hemodialysis; HFD, high-flux dialysis; I/R, ischemia/reperfusion; IL-1β, interleukin-1 beta; IL-6, interleukin 6; IS, indoxyl sulfate; IU, International unit; i.v., intravenous; MDA, malondialdehyde; MitoQ, mitoquinone; OF-IN, oligofructose-enriched inulin; PCS, p-cresyl sulfate; QD, quaque die; RER, resting energy requirement; RCT, randomized controlled trial; SCFA, short-chain fatty acids; SDMA, symmetric dimethylarginine; UA, uric acids; UTs, uremic toxins.

**Table 3 toxins-13-00573-t003:** CAM treatments for the control of UTs.

Intervention	Route, Dosage, and Frequency	Author/Year	Mechanism/Usage	Study Design	Subjects	Subject Number	Result
Clinical Studies							
*Curcuma longa* and *Boswellia serrata*	Oral, 1 capsule, BID	Moreillon et al., 2013 [91]	Anti-inflammation, inhibition of NF-Κb and MAPK	RCT	Patients with CKD	16	IL-6 ↘
UCG	Oral, 5 g, TID and 10 g HS	Zheng et al., 2017 [95]	Anti-inflammation and antifibrosis	RCT	Patients with CKD	292	Cr ↘
Acupuncture	External, LI4, ST36 and KI3, 1 time/week	Yu et al., 2017 [92]	Improving renal local microcirculation	RCT	Patients with CKD	59	Cr ↘
Moxibustion	External, 0.5~7 sessions/week	Zhou et al., 2020 [93]	Dilating local renal capillaries, alleviating kidney podocyte injury	MA	Patients with CKD	1571	BUN ↘, Cr ↘
Animal Studies							
DFD	Gastric gavage, 2.5 g/kg, QD	Tu et al., 2014 [96]	Inhibiting apoptosis by blocking TGF-b1-JNK	Animal	Adenine-induced renal injury rats	27	BUN ↘, Cr ↘, UA ↘
DHI and salvianolic acids	Extracorporeal, DHI 4.16 mL/kg or LA 24.69 mg/kg, once	Li et al., 2019 [97]	Protein-binding competitors	Animal	CKD rats with accumulated IS and pCS	16	Enhanced dialysis removal of IS and pCS
UCG	Gastric gavage, 5 g/kg, QD	Huang et al., 2014 [98]	Antifibrosis, regulation of ECM degradation	Animal	Adenine and UUO-induced renal failure rats	26	BUN ↘, Cr ↘, UA ↘
ZDW	i.p., 2 g/kg, once	Hsu et al., 2014 [99]	Attenuation of apoptosis through limiting of caspase-3 activation	Animal	Gentamicin-induced renal injury rat	12	BUN ↘Cr ↘
ZDW	Embryo exposure, 100 ppm, once	Lu et al., 2020 [100]	Suppression of proinflammatory gene expression	Animal	AA-intoxicated zebrafish embryos	150	tnf-α ↘
Catechin	Oral, 100 mg/kg, QD	Korish et al., 2008 [101]	Antioxidation	Animal	5/6 nephrectomy rats	40	ADMA ↘
Cyanidin-3-O-glucoside (C3G)	i.p., 20 mg/kg, QD	Qin et al., 2018 [102]	Antioxidation	Animal	db/db mice with DN	60	BUN ↘, Cr ↘
EGCG	i.p., 50 mg/kg, QD	Wang et al., 2015 [103]	Anti-inflammation and antioxidation through inhibition of the NF-κB signaling pathway and activation of the Nrf2-Keap1 pathway	Animal	UUO mice	24	BUN ↘, Cr ↘
Gypenoside (GP)	i.v., 50 mg/kg, once	Ye et al., 2016 [104]	Attenuating inflammatory and oxidative stress by inhibiting ERK signaling	Animal	I/R-induced renal injury mice	30	BUN ↘, Cr ↘, IL-1β ↘, IL-6 ↘, MDA ↘, TNF-α ↘
Huangkui capsule	Gastric gavage, 0.75 g/kg, QD	Cai et al., 2017 [105]	Inhibition of the NADPH oxidase/ROS/ERK pathway	Animal	Adenine-induced CRF Rats	18	BUN ↘, Cr ↘
Huangkui capsule	Gastric gavage, 0.675 g/kg, QD	Wang et al., 2019 [106]	Inhibition of the transformation of Trp to indole	Animal	5/6 nephrectomy Rats	21	IS ↘
Leonurine (LEO)	i.v., 50 mg/kg, QD	Xu et al., 2014 [107]	Inhibition of inflammatory and oxidative stress through downregulation of NF-kB	Animal	LPS-induced renal injury mice	120	BUN ↘, Cr ↘, IL-1 ↘, IL-6 ↘, IL-8 ↘, MDA ↘, TNF-α ↘
Ligustrazine (LIG)	i.p., 80 mg/kg, once	Feng et al., 2011 [108]	Downregulation of oxidative stress and apoptosis, decrease in neutrophil infiltration	Animal	I/R-induced renal injury mice	48	MDA ↘, TNF-α ↘
Notoginsenoside R1 (NR1)	i.p., 80 mg/kg, once	Liu et al., 2010 [109]	Blocking apoptosis and inflammatory response by suppressing p38 and NF-kB	Animal	I/R-induced renal injury rats	24	Cr ↘, TNF-α ↘
Osthole	i.p., 40 mg/kg, once	Luo et al., 2016 [110]	Abrogating inflammation by suppressing JAK2/STAT3 signaling, activating PI3K/Akt signaling	Animal	I/R-induced renal injury rats	70	BUN ↘, Cr ↘, IL-6 ↘, TNF-α ↘
Paeoniflorin (PF)	i.p., 30 mg/kg, once	Liu et al., 2015 [111]	Attenuation of inflammatory response by inhibiting CXCR3/CXCL	Animal	ConA-induced renal injury mice	60	BUN ↘, Cr ↘, IL-1β ↘
Resveratrol	Gastric Gavage, 1 mg/kg, QD	Chen et al., 2016 [112]	Modulation of intestinal microbiota	Animal	ApoE(-/-) mice	20	TMAO ↘
Tanshinone I	i.p., 120 mg/kg, QD	Feng et al., 2013 [113]	Enhancement of AAI metabolism by induction of CYP1A	Animal	AAI-induced renal injury mice	40	BUN ↘, Cr ↘
Rhubarb	Enema, 0.5 g, QD	Lu et al., 2015 [114]	Antioxidation, anti-inflammation	Animal	5/6 nephrectomy rats	28	Cr ↘, IS ↘
Rhubarb	Enema, 2.12 g/kg, QD	Ji et al., 2020 [115]	Modulation of intestinal microbiota, improving the intestinal barrier, anti-inflammation	Animal	5/6 nephrectomy rats	30	IL-1β ↘, IL-6 ↘
SkQR1	i.p., 400 nmol/kg, once	Plotnikov et al., 2011 [116]	Antioxidation	Animal	Glycerol-induced rhabdomyolysis rats	36	BUN ↘, MDA ↘

↘, decrease; AA, aristolochic acid; ADMA, asymmetric dimethylarginine; ApoE, apolipoprotein E; BID, bis in die; BUN, blood urea nitrogen; CKD, chronic kidney disease; ConA, concanavalin A; Cr, creatinine; CRF, chronic renal failure; CXCR3/CXCL, c-X-c motif chemokine receptor 3/C-X-C motif Chemokine ligand; CYP1A, cytochrome P450 1A; DFD, Dahuang Fuzi Decoction; DHI, Danhong injection; DN, diabetic nephropathy; ECM, extracellular matrix; EGCG, eepigallocatechin-3-gallate; ERK, extracellular signal-regulated kinase; HS, hora somni; i.p., intraperitoneal injection; I/R, ischemia/reperfusion; IL-1, interleukin-1; IL-1β, interleukin-1 beta; IL-6, interleukin-6; IS, indoxyl sulfate; JAK2/STAT3, Janus kinase 2/signal transducer and activator of transcription 3; LA, lithospermic acid; MA, meta-analysis; MAPK, mitogen-activated protein kinase; MDA, malondialdehyde; NADPH, nicotinamide adenine dinucleotide phosphate; NFκB, nuclear factor kappa B; Nrf2-keap1, nuclear factor E2-related factor 2-Kelch-like ECH-associated protein 1; PCS, p-cresyl sulphate; PI3K/Akt, phosphatidylinositol 3-kinase/Ak strain transforming; QD, quaque die; RCT, randomized controlled trial; ROS, reactive oxygen species; SkQR1, 10-(6′-plastoquinonyl) decylrhodamine; TGF β1-JNK, transforming growth factor-beta-1-c-Jun N-terminal kinase; TID, ter in die; TMAO, trimethylamine N-oxide; TNF-α, tumor necrosis factor alpha; UA, uric acid; UCG, uremic clearance granule ; UUO, unilateral ureteral obstruction; ZDW, Zhibai Dihuang Wan.

## Data Availability

The data used to support the findings of this study are included within the article.

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
