# Peer review of "The Therapeutic Strategies for Uremic Toxins Control in Chronic Kidney Disease"

_toxins, 2021, doi:10.3390/toxins13080573_

Round 1

Reviewer 1 Report

Intestinal Catabolism by microbiota is a rich source of nutritional supplement, potentially overcoming genetic limitation human genome, adding multiple additional enzymes to create micro-nutrients by bacterial microbiome.

However, this microbiota is also a rich source of uremic toxin and GI absorption +/- liver conversion is the predominant source. This paper is attempting to describe novel potential extrarenal avenues to decrease urinemic toxin formation and/or facilitation disposal from human body

Overall further Englsih editing still desirable

Detailed comments.

Section 2 is much improved – may wish to replace “Western” medicine with “Classic  (or conventional) Medical Therapies

Split of Table (Table 1., 2., 3.) to human/animal studies is well doen, make resulst easier to read on this difficult subject

Minor comments:

-plants names should Italicized

-some abbreviations still not explained: e.g., SKQR1

Section 4.21.: moxibustion still not well explained

Suggested addition to the literature to discuss:

PMID: 31797710

PMID: 29055386

Author Response

Reviewer1

Intestinal Catabolism by microbiota is a rich source of nutritional supplement, potentially overcoming genetic limitation human genome, adding multiple additional enzymes to create micro-nutrients by bacterial microbiome.

However, this microbiota is also a rich source of uremic toxin and GI absorption +/- liver conversion is the predominant source. This paper is attempting to describe novel potential extrarenal avenues to decrease urinemic toxin formation and/or facilitation disposal from human body

Overall further Englsih editing still desirable

Detailed comments.

Comment 1: Section 2 is much improved – may wish to replace “Western” medicine with “Classic  (or conventional) Medical Therapies

Response 1: Thank you very much for your comments.

We have replaced “Western” medical therapies with conventional medical therapies.

(Please see p1, L28; p3, L103-104)

Comment 2: Split of Table (Table 1., 2., 3.) to human/animal studies is well doen, make resulst easier to read on this difficult subject

Response 2: Thank you very much for your comments. 

Minor comments:

Comment 3: -plants names should Italicized

Response 3: Thank you very much for your comments. 

We have changed the form of plant Latin name to italics.

(Please see p12, L395-p16, L558)

Comment 4: -some abbreviations still not explained: e.g., SKQR1

Response 4: Thank you very much for your comments. 

We have explained abbreviations in the article.

(Please see p15, L550-551)

Comment 5: Section 4.21.: moxibustion still not well explained

Response 5: Thank you very much for your comments. 

We have supplemented more information about moxibustion.

(Please see p16, L577-582)

Comment 6: Suggested addition to the literature to discuss:

PMID: 31797710

PMID: 29055386

Response 6: Thank you very much for your comments. 

We have added the important papers you suggested to the discussion

(Please see p2, L84-89, p5, L162-164)

Reviewer 2 Report

I congratulate the authors for the huge work done in this complex research field. 

My comments are written here:

Line 5, Change microbiome to microbiota, microbiome represent the whole microbe genome,  Microbiota is usually defined as the assemblage of living microorganisms present in a defined environment (e.g. intestinal environment).

Lines 53, 58, 108 and subsequent. Do not use 'flora' instead of 'microbiota', is too colloquial.

After the introduction. Insert the method used to search and screen the articles, such as the database and keywords.

In the 3 tables insert a column with the number of subjects or animals analyzed in the study.

Section 2.10, lines 202-206, details and explains the poor bioavailability of reduced glutathione in oral supplements.

Table 2 specifies the probiotic name used in the study [49] and synbiotic in the study [52].
For study [51] specify the type of SCFA used. 
For vegetarian intervention insert the studies and not the narrative review.

Line 225 insert the alpha-linoleic acid acronym (ALA).

Line 257, insert the more detailed definition of probiotics, not only ''live microorganism".

Line 262, specify the formulation of Kibow Biotics.

Line 265, 296 and subsequent. The name of probiotics must be written in italics.

Line 281, O3FAs cannot derive from mineral oils, maybe from vegetable oil.

Line 334, please substitute Herbal monomers with the herbal active principle.

Line 353, 356, 358 and subsequent. Change format of the Latin name of plants in italics.

In the TCM preparation, insert the Latin name of the plants and not only the TCM name.

Line 411 and subsequent, In vitro and in vivo, must be written in italics.

Line 486,487 specify what is the 'certain type of microbiota'.

Good work!

Author Response

Reviewer 2

I congratulate the authors for the huge work done in this complex research field. 

My comments are written here:

Comment 1: Line 5, Change microbiome to microbiota, microbiome represent the whole microbe genome,  Microbiota is usually defined as the assemblage of living microorganisms present in a defined environment (e.g. intestinal environment).

Response 1: Thank you very much for your comments. 

We have changed the microbiome to microbiota.

(Please see p1, L15)

Comment 2: Lines 53, 58, 108 and subsequent. Do not use 'flora' instead of 'microbiota', is too colloquial.

Response 2: Thank you very much for your comments. 

We have changed the flora to microbiota.

(Please see p2, L63, L68, p7, L243)

Comment 3: After the introduction. Insert the method used to search and screen the articles, such as the database and keywords.

Response 3: Thank you very much for your comments. 

We have added the search method after the introduction.

(Please see p3, L95-98)

Comment 4: In the 3 tables insert a column with the number of subjects or animals analyzed in the study.

Response 4: Thank you very much for your comments. 

We have added a column with the number of subjects analyzed in the study.

(Please see table 1, 2 and 3)

Comment 5: Section 2.10, lines 202-206, details and explains the poor bioavailability of reduced glutathione in oral supplements.

Response 5: Thank you very much for your comments. We have extensively revised our manuscript and explained the poor bioavailability of reduced glutathione in oral supplements according to your request. 

(Please see table 1, 2 and 3)

(Please see p7, L225-240)

Comment 6: Table 2 specifies the probiotic name used in the study [49] and synbiotic in the study [52].

Response 6: Thank you very much for your comments. 

We have added the name of probiotic and symbiotic used in table 2.

(Please see table 2)

Comment 7: For study [51] specify the type of SCFA used. 

Response 7: Thank you very much for your comments. 

We have added the type of SCFA used

(Please see table 2)

Comment 8: For vegetarian intervention insert the studies and not the narrative review.

Response 8: Thank you very much for your comments. 

We have used cohort study instead of narrative review.

(Please see table 2)

(Please see p10, L349-350)

Comment 9: Line 225 insert the alpha-linoleic acid acronym (ALA).

Response 9: Thank you very much for your comments. 

We have added the acronym of alpha-linoleic acid.

(Please see p8, L260)

Comment 10: Line 257, insert the more detailed definition of probiotics, not only ''live microorganism".

Response 10: Thank you very much for your comments. 

We have supplemented more information about probiotics.

(Please see p9, L295-299)

Comment 11: Line 262, specify the formulation of Kibow Biotics.

Response 11: Thank you very much for your comments. 

We have supplemented the formulation of Kibow Biotics.

(Please see p9, L302-303)

Comment 12: Line 265, 296 and subsequent. The name of probiotics must be written in italics.

Response 12: Thank you very much for your comments. 

We have changed the form of probiotics name to italics.

(Please see p9, L284, p10, L336)

Comment 13: Line 281, O3FAs cannot derive from mineral oils, maybe from vegetable oil.

Response 13: Thank you very much for your comments. 

We have corrected the mistake and changed the mineral oils to vegetable seed oils.

(Please see p9, L321)

Comment 14: Line 334, please substitute Herbal monomers with the herbal active principle.

Response 14: Thank you very much for your comments. 

We have substitute herbal monomers with the herbal active principle.

(Please see p10, L376)

Comment 15: Line 353, 356, 358 and subsequent. Change format of the Latin name of plants in italics.

Response 15: Thank you very much for your comments. 

We have changed the form of plant Latin name to italics.

(Please see section 4)

Comment 16: In the TCM preparation, insert the Latin name of the plants and not only the TCM name.

Response 16: Thank you very much for your comments. 

We have insert the Latin name of the plants in the TCM preparation

(Please see section 4.4, 4.5)

Comment 17: Line 411 and subsequent, In vitro and in vivo, must be written in italics.

Response 17: Thank you very much for your comments. 

We have changed the form of In vitro and in vivo to italics in the article.

(Please see p13, L456, L481, L483)

Comment 18: Line 486,487 specify what is the 'certain type of microbiota'.

Good work!

Response 18: Thank you very much for your comments. 

We have specified the 'certain type of microbiota'.

(Please see p15, L532)

This manuscript is a resubmission of an earlier submission. The following is a list of the peer review reports and author responses from that submission.

Round 1

Reviewer 1 Report

Congratulations, you did a great job.

The article is large and important work for the scientific community. 

CAM therapy has small clinical data and only hypothesis from an animal study. 

Author Response

Reviewer 1 

Comment 1: Congratulations, you did a great job. The article is large and important work for the scientific community. CAM therapy has small clinical data and only hypothesis from an animal study.

Response 1: Thank you very much for your comments. The clinical data of high-quality CAM therapy is indeed scarce. Prescriptions and acupuncture have been studied in RCT grade, and the evidence of moxibustion belongs to meta-analysis grade. However, most of the CAM therapy studies are still animal experiments. Fortunately, it is indeed observed that CAM therapy has the effect of reducing urinary toxins and improving renal function in the animal experiment.

Reviewer 2 Report

The submitted paper has potential but far from acceptable in the current format. The Authors are approaching a huge and hugely important subject; however, their review is not thoroughly done and the “bulleted” description of each presumed anti-uremic substances is rather superficial… in this current write-up, it is merely a listing of various substances with little deeper thinking and making connection, to make a full use for a potential of a review paper

Little efforts have been created to offer even a glimpse of potential ‘quantitate’ comparison between these approaches/agents

Human/anemia models are widely mixed together, as if they would have the same quantitative weight

The Chinese CAM approaches should be viewed and described with particular scrutiny, as this paper presents to a world-wide audience. Do not assume any pre-existing knowledge of the reader about any of these!

  1. Decoctions are particularly problematic a they tend to mixtures
  2. UCG very poorly explained

Just to mention a few examples
1. How dose allopurinol got into this topic? Very difficult to see how this would fit into this conceptual collection…

  1. AST-120 write-up is particularly weak
  2. some abbreviation brought in w/o explaining (e.g., SCFA, SKQR1)

And many mores..

Notice: ‘Authors’ contributions” and “Informed Consent statement’ are not even properly completed or developed

Additional/conceptual problems:

The Authors fail to fully utilize a deeper understanding constipation prolong gut-stool (bacteria) interaction, thus the potential for larger absorption of uremic substances. They fail to fully connect the dots between liver conversion of non-water soluble uremic toxins – these may not be excreted in the urine but with bile (and hence, potential for additional damage to GI barrier). The failure of discussing (a related subject)  why peritoneal dialysis would work so well in ESRD – as it preferentially dialysis the very compartment generating the predominance of uremic toxins (GI compartment). The potential for vegan diet to represent dietary alkali delivery not even considered yet

Author Response

Response to reviewer’s comments

Reviewer 2

The submitted paper has potential but far from acceptable in the current format. The Authors are approaching a huge and hugely important subject; however, their review is not thoroughly done and the “bulleted” description of each presumed anti-uremic substances is rather superficial… in this current write-up, it is merely a listing of various substances with little deeper thinking and making connection, to make a full use for a potential of a review paper.

Comment 1: Little efforts have been created to offer even a glimpse of potential ‘quantitate’ comparison between these approaches/agents

Response 1: Thank you very much for your comments. Sorry for neglecting the dosage of treatments. We have added the dosage and frequency of various treatments for comparison in table.

(Please see table 1, 2 and 3)

Comment 2: Human/anemia models are widely mixed together, as if they would have the same quantitative weight

Response 2: Thank you very much for your comments. Human and animal should have the different quantitative weight. According to your suggestion, in order to facilitate the reading and clearly separate clinical research and animal experiments, we have distinguished human and animal studies in the table.

(Please see table 1, 2 and 3)

Comment 3: The Chinese CAM approaches should be viewed and described with particular scrutiny, as this paper presents to a world-wide audience. Do not assume any pre-existing knowledge of the reader about any of these!

Response 3: Thank you very much for your comments. We have added more description of CAM.

(Please see p9, L296-311)

Comment 4: Decoctions are particularly problematic a they tend to mixtures

Response 4: Thank you very much for your comments. The clinical data of high-quality CAM therapy is indeed scarce. A meta-analysis including 20 RCTs showed that in patients with diabetic kidney dis-ease (DKD), traditional Chinese medicine (TCM) as an adjunctive treatment significantly reduced BUN and serum creatinine levels, and improved uremic protein excre-tion and quality of life compared to placebo group with less adverse events. (Zhang, L.et al. Chinese herbal medicine for diabetic kidney disease: a systematic review and meta-analysis of randomised placebo-controlled trials. BMJ open 2019, 9, e025653.) Decoctions therapy for UT control in CKD has been studied in RCT grade. However, most of the CAM therapy studies are still animal experiments. Fortunately, it is indeed observed that CAM therapy has the effect of reducing urinary toxins and improving renal function in the animal experiments. For CKD, finding new treatment targets and methods, and understanding the efficacy mechanism of decoctions, herbs, and even single pure compound one by one has always been the goal of CAM scholars and pharmacologists.

(Please see p9, L296-311 and table 3)

Comment 5: UCG very poorly explained

Response 5: Thank you very much for your comments. We have supplemented more information about UCG.

(Please see p11, L348-360 and table 3)

Comment 6: How dose allopurinol got into this topic? Very difficult to see how this would fit into this conceptual collection…

Response 6: Thank you very much for your comments. Uric acid (UA) belongs to the small water-soluble urinary toxin according to the study published by Vanholder in 2003 (Vanholder et al. New insights in uremic toxins. Kidney Int Suppl 2003, 10.1046/j.1523-1755.63.s84.43.x, S6-10). Furthermore, a randomized controlled trial reported that the administration of allopurinol in patients with CKD for 24 months reduced the serum UA level by inhibiting UA synthesis and reduce the C-reactive protein level. So we collect the allopurinol into our article. According to your suggestion, we deleted the description of allopurinol.

Comment 7: AST-120 write-up is particularly weak

Response 7: Thank you very much for your comments. We have supplemented more information about AST-120.

(Please see p4, L119-p5, L143)

Comment 8: some abbreviation brought in w/o explaining (e.g., SCFA, SKQR1)

And many mores..

Response 8: Thank you very much for your comments. Although the full name of SCFA has appeared in the previous paragraph, it should be explained in the subtitle as you suggested. In addition, SKQR1 is the name of a synthetic antioxidant that targets mitochondria, not an abbreviation. We have checked the abbreviations in the article and explain them.

(Please see p8, L263)

Comment 9: Notice: ‘Authors’ contributions” and “Informed Consent statement’ are not even properly completed or developed

Response 9: Thank you very much for your comments. Sorry for the incomplete information due to the conversion of the manuscript during the submission process. We have added the missing information in the article.

(Please see p14, L490-503)

Comment 10: Additional/conceptual problems:

The Authors fail to fully utilize a deeper understanding constipation prolong gut-stool (bacteria) interaction, thus the potential for larger absorption of uremic substances. They fail to fully connect the dots between liver conversion of non-water soluble uremic toxins – these may not be excreted in the urine but with bile (and hence, potential for additional damage to GI barrier).

Response 10: Thank you very much for your comments. The article describes the classification of urinary toxins, we know many hundreds of UTs including small water-soluble solutes, which can be removed through dialysis, and larger and protein-bound molecules, which are less likely to be removed during dialysis. Furthermore, the article described the possible mechanism of uremic toxin generation, and pointed out that the larger and protein-bound molecules that have not been dialyzed will be converted into uremic toxin through the metabolism of intestinal bacteria and the liver enzyme, thereby destroying the protective barrier of the intestinal epithelium, leading to the typical destruction of normal metabolic balance and uremic homeostasis that results in inflammation and uremia, causing multiple organ damage. We have added the effect of constipation on changes in intestinal flora and the accumulation of urinary toxins followed your expert suggestion (Sumida, K.; Yamagata, K.; Kovesdy, C.P. Constipation in CKD. Kidney Int Rep 2020, 5, 121-134).

(Please see p2, L62-66)

Comment 11: Additional/conceptual problems:

The failure of discussing (a related subject) why peritoneal dialysis would work so well in ESRD – as it preferentially dialysis the very compartment generating the predominance of uremic toxins (GI compartment).

Response 11: Thanks for your kind reminds. According to the review articles, due to the use of a highly permeable membrane with a greater pores radius and better preservation of the residual renal function, peritoneal dialysis (PD) could be anticipated that some uremic toxins are more efficiently cleared across the peritoneal membrane, and that the plasma levels of p-Cresol (protein-bound uremic toxin) are lower than in hemodialysis patients. (Lameire, N.; Vanholder, R.; De Smet, R. Uremic toxins and peritoneal dialysis. Kidney Int Suppl 2001, 78, S292-297 and Dhondt, A.; Vanholder, R.; Van Biesen, W.; Lameire, N. The removal of uremic toxins. Kidney Int Suppl 2000, 76, S47-59). However, our article focuses on non-dialysis therapies for uremic toxins. This is why we had not addressed this viewpoints in our manuscript.

Comment 12: The potential for vegan diet to represent dietary alkali delivery not even considered yet

Response 12: Thanks for your kind reminds. In fact, plants are the dietary source of fibers, which not only shifts the gut microbiota to increased generation of anti-inflammatory compounds and but also reduced producing of uremic toxins. Moreover, vegatarian diets have low endogenous acid load, which could reduce metabolic acidosis in CKD patients and potentially slow CKD progression [Chauveau et al. Nephrol Dial Transplant (2019) 34: 199–207 and Carrero et al. Nat Rev Nephrol. 2020 Sep;16(9):525-542]. According to your suggestions, we had added these statements in 3.9 Vegatarian diet.

(Please see p8, L279-p9, L288)
